# Effectiveness of Biocidal Paint Containing Permethrin, Ultramarine and Violet 23 Against *Alphitobius diaperinus* (Panzer) (Coleoptera: Tenebrionidae) in Laboratories and Poultry Houses

**DOI:** 10.3390/ani10091461

**Published:** 2020-08-20

**Authors:** Sara Dzik, Tomasz Mituniewicz

**Affiliations:** Department of Animal and Environmental Hygiene, Faculty of Animal Bioengineering, University of Warmia and Mazury in Olsztyn, 5 Oczapowski Street, 10-719 Olsztyn, Poland; t.mituniewicz@uwm.edu.pl

**Keywords:** lesser mealworm, insect pest, poultry, broiler houses, insecticide, repellent, limewash

## Abstract

**Simple Summary:**

*Alphitobius diaperinus* (Panzer) (Coleoptera: Tenebrionidae) is an insect pest commonly found in poultry houses. High air temperature that accompanies chicken rearing, high relative air humidity and deposited chicken excreta are the factors that create favorable conditions for the development of *A. diaperinus* on poultry farms throughout the year. *A. diaperinus* is a vector for pathogenic bacteria (including *Salmonella* spp., *Campylobacter* spp., *Staphylococcus* spp., *Escherichia coli*), fungi (*Aspergillus* spp., *Candida* spp.), protozoa (*Eimeria* spp.), poultry parasites and viruses (causing Marek’s disease, Gumboro disease, Newcastle disease, avian influenza, and leukemia in poultry). At the same time, the insect’s resistance to commonly used insecticides is observed. Therefore, it is crucial to find an effective solution to eliminate the insect. In view of the above, an attempt to reduce the population of *A. diaperinus* on poultry houses is the justification for this research. In the research we used biocidal paint containing permethrin and a mixture of ultramarine and violet 23 to control the number of *A. diaperinus* (larvae and adults) in laboratories and poultry houses.

**Abstract:**

Reducing *Alphitobius diaperinus* in poultry production is a difficult task. However, attempts should be made to control the insect pest, as it poses a serious threat to the life and health of the chickens, as well as the workers on a farm. Our research was conducted in two stages to assess the effectiveness of the biocidal paint against *A. diaperinus*, containing active substances such as permethrin and a mixture of ultramarine and violet 23. In the first stage, under laboratory conditions, after 22 days, 100% mortality of *A. diaperinus* larvae and adults was achieved. This allowed us to assume that the biocidal paint may also be effective in poultry houses. In the poultry house where biocidal paint was applied, the number of insects decreased continuously alongside the sampling dates. In both research stages, the biocidal paint proved more effective against *A. diaperinus* than traditional limewash, and also the time to effective interaction of the paint was noted. Additionally, it was observed that the larvae were more susceptible to the active substances than adults. The research was practical, however, further analyses are necessary to fully control *A. diaperinus*, especially in poultry houses.

## 1. Introduction

In scientific research, an increased interest in *Alphitobius diaperinus* (Panzer) (Coleoptera: Tenebrionidae) was noticeable at the end of the 20th century [1,2,3,4,5,6]. This resulted from the growing number of poultry farms which emerged as an answer to an increased demand for poultry meat on the food market. *A. diaperinus* (commonly called the lesser mealworm, the darkling beetle or litter beetle) is one of the factors that generates significant losses to farmers in present day [7,8,9,10]. The insect is mostly found in poultry houses [7,11,12,13], especially when the litter is slightly moist [14]. The insects cause a lot of damage by tunneling in the crevices in floors and walls, thus destructing insulation [8,11]. According to Singh [15], energy costs in beetle damaged broiler houses are reported to be 67% higher than in houses without beetle damage. The damage mainly occurs when the larvae are seeking places to pupate. The problem of over-population of *A. diaperinus* is not limited to Central Europe, but has spread other regions of the world [16,17,18,19,20,21].

Additionally, the birds that consume the feed that contains the insects [22] may suffer from so-called ‘nutritional stress’, since birds in general have a limited chitinolytic activity in the digestive tract (due to the deficiency of the enzyme responsible for digesting chitin which covers the insects’ body) [23,24,25]. As reported by Khempka et al. [26], the digestibility of chitin in broiler chickens is at the level of 18–24%, which may lead to an increased secretion of digestive juices, increased appetite, a decrease in the pH of the digestive tract and intestinal obstruction [15].

It has been experimentally proved that *A. diaperinus* is also a vector for many serious diseases of both poultry and humans [8,9,10]. The insects become infected by feeding on bird feces and dead chickens [7], and may remain infectious for a long period of time. The poultry become infected with pathogens while consuming the infected larvae or adult insects. As reported by Hess et al. [8], Yeasmin et al. [10], Singh and Johnson [27], and Crippen et al. [28], *A. diaperinus* is a carrier of the dangerous pathogen of *Salmonella*, *Campylobacter* spp. and *Escherichia coli* [9,29]. Additionally, it is a vector for a virus which causes Marek’s disease and avian leukosis [9].

Extensive laboratory research is being conducted in order to find solutions to effectively reduce *A. diaperinus* number. The most significant studies investigate the influence of different groups of insecticides and bioactive compounds on the insect [30]. To achieve this, it is crucial to analyze the insect’s behavior, and not only in laboratories but also in poultry houses [11]. A tendency of *A. diaperinus* to increase its resistance to the applied insecticides has been observed [16]. The resistance of *A. diaperinus* against the most common insecticides (cypermethrin, dichlorophen, triflumuron) was tested in poultry houses in Brazil. The highest susceptibility of the insects to cypermethrin and dichlorophen was observed in these poultry houses where the compounds had not been used for two years prior to the research. However, the application of cypermethrin at each washing and the disinfection of the poultry houses resulted in high resistance of *A. diaperinus* to this pyrethroid. Furthermore, it has been suggested that there exists cross-resistance between cypermethrin and dichlorophen, so it is crucial to use different methods to control *A. diaperinus* (including alternatives to chemicals) in order to reduce the likelihood of the increase in the insect’s resistance to insecticides [16].

Therefore, it is important to find a universal and effective solution to control *A. diaperinus* [27]. Paints with insecticidal properties are known, but until now they have not been used in poultry buildings against *A. diaperinus*. A new technique is biocidal paint (Kleib Ltd., Brzesc Kujawski, Poland) containing chemical and optical repellent. The paint contains active substances: permethrin, ultramarine and violet 23. Permethrin is a synthetic pyrethroid insecticide, commonly used against many insects (chemical repellent) [31,32,33]. The insecticide affects the insect’s nervous system by paralyzing—muscle contraction, paralysis and then death can occur as a result. Permethrin acts as a stomach poison when eaten by insects or as a contact poison through direct contact with target pests. It is effective against eggs, larvae, pupae and adults [34]. In addition, some beetles have photoreceptors sensitive to blue or violet light [35]. By adding blue pigment (ultramarine) and violet pigment (violet 23) to the paint, it can be an effective agent in the fight against insects (optical repellent) [33]. In view of the above, an attempt to reduce the population of *A. diaperinus* in livestock buildings for poultry is the justification for this research.

The aim of this study was to assess the effectiveness of an insecticide in the form of biocidal paint on *A. diaperinus* in laboratories and poultry houses. Thereby, the effectiveness of the biocidal paint was compared with limewash—a common method used against insect pests in Poland. It has been assumed that the application of the biocidal paint will reduce the population of *A. diaperinus*, both larvae and adults. Moreover, we assumed that over time the active substances present in the paint will systematically reduce the number of insect pests (a long-term insecticide method), and that the biocidal paint will prove more effective against *A. diaperinus* than limewash.

## 2. Materials and Methods

In order to assess the effectiveness of the biocidal paint, the experiment was divided into two stages: in laboratories and poultry houses.

### 2.1. Experimental Factor in Experiments

The biocidal paint is a white, ecological, biodegradable biocidal paint for livestock buildings, with exceptionally low odor emission. The paint received a hygienic certificate issued by the National Hygiene Institute in Warsaw, Poland (No. HK/B/0571/01/2016) and a patent issued by the Polish Patent Office (No. 232978). The biocidal paint contains two insect repellents:(i)chemical—permethrin—C_21_H_20_C_l2_O_3_—3-phenoxyphenylomethyl 3-(2,2-dichloroethenyl)-2,2-dimethylcyclopropane-1-carboxylate;(ii)optical—a mixture of ultramarine and violet 23—A_l6_Na_6_O_24_S_8_Si_6_—hexasodium; bis(2,4,5-trioxa-1-sila-3-aluminabicyclopentan-1-yloxy)alumanyloxy-dioxido-2,4,5-trioxa-1-sila-3-aluminabicyclopentan-1-yloxy(trioxidosilyloxy)alumanyloxysilane; 1-oxido-2,4,5-trioxa-1-sila-3-aluminabicyclopentane;tetrathietane.

### 2.2. Laboratory Bioassay

To evaluate the influence of the biocidal paint on *A. diaperinus*, initially laboratory tests were conducted. *A. diaperinus* (larvae and adults) were collected from a poultry farm located in Pomerania, northern Poland, where limewash was used against insect pests. The beetles were collected from drinkers and feeders [36] together with litter. Having been transported to the laboratory, the insects were kept for 7 days in plastic bags before the experiment began [21,36]. According to Wolf et al. [21], later on, beetles were kept in snap-on boxes measuring 30 cm × 20 cm × 20 cm and each box was padded with collected litter (~0.003 m^3^). The boxes were plexiglass, covered with sealing foam (to reflect the conditions of the poultry houses on the farm). The insects were kept at 22 °C ± 2 °C and 60–70% relative humidity [37] and fed on standard industrial broiler chicken feed. Water was provided with a moistened sponge of 6 cm^2^ [36,38]. The interiors of the boxes were painted with a brush using (i) limewash—control group (hydrated lime + water) and: (ii) biocidal paint—experimental group. Insect mortality as well as mortality difference between larvae and adults were assessed after 7, 10, 14 and 22 days. The insects were removed from the boxes with surgical tweezers, placed on a Petri dish and counted [21]. The insects were considered dead when they did not move having been touched with surgical tweezers. The experiment was repeated 3 times over time—18 boxes and two groups were used (experimental scheme—Table 1). Each box was supplemented with 50 insects (divided into 25 adults and 25 larvae), totaling 900 individuals: 450 larvae and 450 adults.

### 2.3. Poultry Bioassay

The research was continued on a poultry farm located in Pomerania, northern Poland. The experiments covered two poultry houses. Each group comprised a poultry house with a maximum stocking density (20,500 birds, Ross 308). The chickens were reared on rye straw litter about 15 cm thick, without padding, for the period of 6 weeks (40 days). Each rearing period was followed by a cleaning period of 3 weeks (after the birds’ removal). Furthermore, no organophosphates or pyrethroids had been used on the farm (in either building) for the period of five years prior to the experiment.

Different methods against *A. diaperinus* were used in the buildings. The poultry house where limewash was applied was the control (C). Limewash was applied after the completion of each chicken flock with a hydrodynamic aggregate after washing, fogging the building, the walls and floor with limewash (300 mL/m^2^*).* In the experimental poultry house (E), biocidal paint was used (400 mL/m^2^; application with a hydrodynamic aggregate). The biocidal paint was applied only once, at the beginning of the experiment. The experiment lasted over one year (423 days). In this time, three treatments were performed (Table 2).

Tenedrop traps for insect pests (Biomucha Company, Bonin, Poland) [39] were used to sample larvae and adults of *A. diaperinus*. The samples were collected after each change of chicken feed. The traps were distributed in the buildings on the 1st day of chicken rearing—before the arrival of the birds. The initial population of *A. diaperinus* in each treatment was considered to be at zero level. The dates of sampling are provided in Table 3. According to Salin et al. [40], in each building 15 traps were placed. The traps were located near the walls and between the feed lines. The traps were set according to Figure 1. The first sampling was performed 3 weeks after the application of the biocidal paint. The trapped insects were put into sealed plastic bags and frozen for 48 h [40,41]. Next, beetles from each trap were placed on a Petri dish and counted [40]. The results obtained in the building with limewash were taken as a baseline estimate of *A. diaperinus* population.

### 2.4. Statistical Analysis

Statistical analysis was conducted with Statistica v. 13.3.0 (TIBCO Software Inc., Palo Alto, CA, USA). Under laboratory bioassay, the groups were compared using the Student’s t-test, and multiple comparisons were analyzed statistically by one-way analysis of variance with repeated-measures (ANOVA) followed by the Tukey test. Under poultry bioassay, data were not normalizable. The analyses were performed using the non-parametric Mann-Whitney U test and evaluated with Friedman analysis of variance (ANOVA) followed by Dunn’s test. For all the tests, *p* < 0.05 was applied. Data distribution normality was verified with the Shapiro-Wilk test [42].

## 3. Results

### 3.1. Laboratory Bioassay

In laboratory bioassay, the effectiveness of the biocidal paint against *A. diaperinus* (larvae and adults) was tested. Furthermore, the effectiveness of the biocidal paint was compared with that of limewash. Different results were observed for the biocidal paint and limewash. The application of the biocidal paint resulted in a significantly higher number of dead larvae and adults, compared to limewash, which proves an improved insecticidal efficacy of this treatment. At the end of the experiment, the mortality rate of *A. diaperinus* (both larvae and adults) reached 100% in the case of the biocidal paint—all the insects were dead.

Control over *A. diaperinus* larvae and adults may be obtained by applying the biocidal paint. Populations of insect pests were reduced by the biocidal paint in all trials (Table 4). Statistical differences in the mortality of the larvae between the control group and the experimental group were observed on days 7, 10, 14 and 22. However, in the case of adults it was noted on days 10, 14 and 22, and no significant differences were observed between the groups on day 7. Additionally, statistical interaction between the biocidal paint and time was noted, which was statistically significant (*p* < 0.05). No statistical interaction between limewash and time was observed, with mortality ranging from 24% to 49.3% for dead larvae and from 32% to 42.7% for dead adults. Biocidal paint presented positive interaction with time against larvae (ranging from 41.3% to 100% after 22 days of exposition). The same was observed for adults. Furthermore, the mortality of the larvae caused by biocidal paint after 7, 10 and 14 days of the experiment was higher than that of the adults. In the case of limewash, the mortality of the larvae was lower in 7 and 10 day of experiment, however, after 14 and 22 days it was higher than that of the adults.

### 3.2. Poultry Bioassay

Biocidal paint has a positive impact on the number of *A. diaperinus*. In the group E, where biocidal paint was used, the number of insect pests decreased in each treatment from about 10,000 to 4200 (Figure 2). It was observed that at the beginning, the number of insects decreased (from treatment I to treatment II it was over 40%). However, in case of treatment II and treatment III the number of *A. diaperinus* was reduced by about 25%. The number of larvae was reduced in treatment II and treatment III in relation to the initial population from 4111 to 1420 (reduction of 66%) and 965 (reduction of 77%). In the case of adults, the reduction was from 5964 to 4249 (29%) and 3274 (45%), respectively. No statistical differences were observed between treatment II and treatment III in the cases of both larvae and adults. In the group C, where limewash was used, the number of insects in treatment I was 8662, in treatment II 14,490 (the number of insects increased by almost 70%) and in treatment III 9045 (an increase of almost 5% compared to the initial population), with no statistical differences between treatment I and treatment III in the number of larvae and adults. Additionally, the number of *A. diaperinus* was higher in treatment II in relation to the population of the beetles in treatment I and treatment III. It was most probably caused by a failure of the ventilation system that lasted for several days. The fault resulted in increased temperature and relative air humidity in the building, which may have caused more favorable living conditions for the insects. Moreover, the population of *A. diaperinus* is subject to seasonality, which translates into an increase in the number of the beetles in the summer.

The effectiveness of the biocidal paint over time was significant (Table 5). The interaction between the biocidal paint and time of treatment was statistically verified. The population of the insects per trap decreased in every treatment. Over five months after the application of the biocidal paint, the maximum number of insects per trap decreased by about 22.86%, and over one year after the application – by about 45.91%. In the case of limewash, the population of the insects increased by 98.40% and later by about 53.97% compared to the initial value. There was no interaction between limewash and time.

## 4. Discussion

The obtained results allowed to assume that the biocidal paint was effective in laboratories and poultry houses and could prove competitive against the commonly used limewash. Over time, since the first application of the biocidal paint, the released bioactive substances reduced the population of *A. diaperinus*. The treatment with the biocidal paint provided long-term control of the insects, lasting over one year (423 days) in poultry houses. Our results allow us to assume that the biocidal paint has a more significant impact on the *A. diaperinus* population throughout the study period than limewash.

The choice of the right product to control *A. diaperinus* in poultry production can be problematic [36,40], mostly regarding the fact that insecticides can only be applied during the cleaning period when there are no birds in the building [20,43]. Some products are not always efficient, as they may not reach the hideouts of the insects, for instance in wall and floor crevices [44]. According to Hickmann et al. [45], currently the most commonly used chemical insecticides are pyrethroids and organophosphates, which, as research has shown, leads to the growing resistance of *A. diaperinus* to these compounds. The resistance of the insect to fenitrothion and permethrin was observed in Great Britain back in 1996. In turn, in Australia, the resistance of *A. diaperinus* against fenitrothion, deltamethrin and cyfluthrin has been experimentally confirmed, whereas in North and South America its resistance against carbaryl, methoxychlor, dichlorodiphenyltrichloroethane, cyfluthrin, permethrin, cypermethrin and chlorpyrifos has been recorded. In Brazil, a very low effectiveness of products containing cypermethrin was observed [45]. Regular identification of insecticidal susceptibility would allow manufacturers to rotate chemicals in order to reduce the insect’s resistance to insecticides in a given population [36,46,47].

Some previous research results already suggested that control of *A. diaperinus* by adding hydrated lime to litter can be obtained due to lime’s hygroscopic characteristics, desiccation, and alkalization [48]. Using chemicals alone to control *A. diaperinus* has proved ineffective [40] in controlling the insect.

Perhaps the combination method would prove effective: applying natural insecticidal methods with a simultaneous reduction in chemical insecticides could prevent the growing resistance of the insects to the active substances. Additionally, the research must be conducted not only under laboratory bioassay but also poultry bioassay, where the microclimate changes and the insect’s behavioral patterns can be observed in its natural habitat. Furthermore, taking into account the sex of the insect may prove effective in controlling it. This is important because it would be worthwhile to examine whether the susceptibility to insecticides is different for males and females.

## 5. Conclusions

The analyzed biocidal paint proved more effective against *A. diaperinus* than commonly used limewash and was more effective against the larvae than adults in laboratories and poultry houses. Furthermore, the method is long-term (in that paint will systematically reduce the number of insect pests) and less labor-intensive than limewash. Additionally, the research is of a practical nature. The results may be very useful for poultry producers in controlling the *A. diaperinus* population.

## Figures and Tables

**Figure 1 animals-10-01461-f001:**
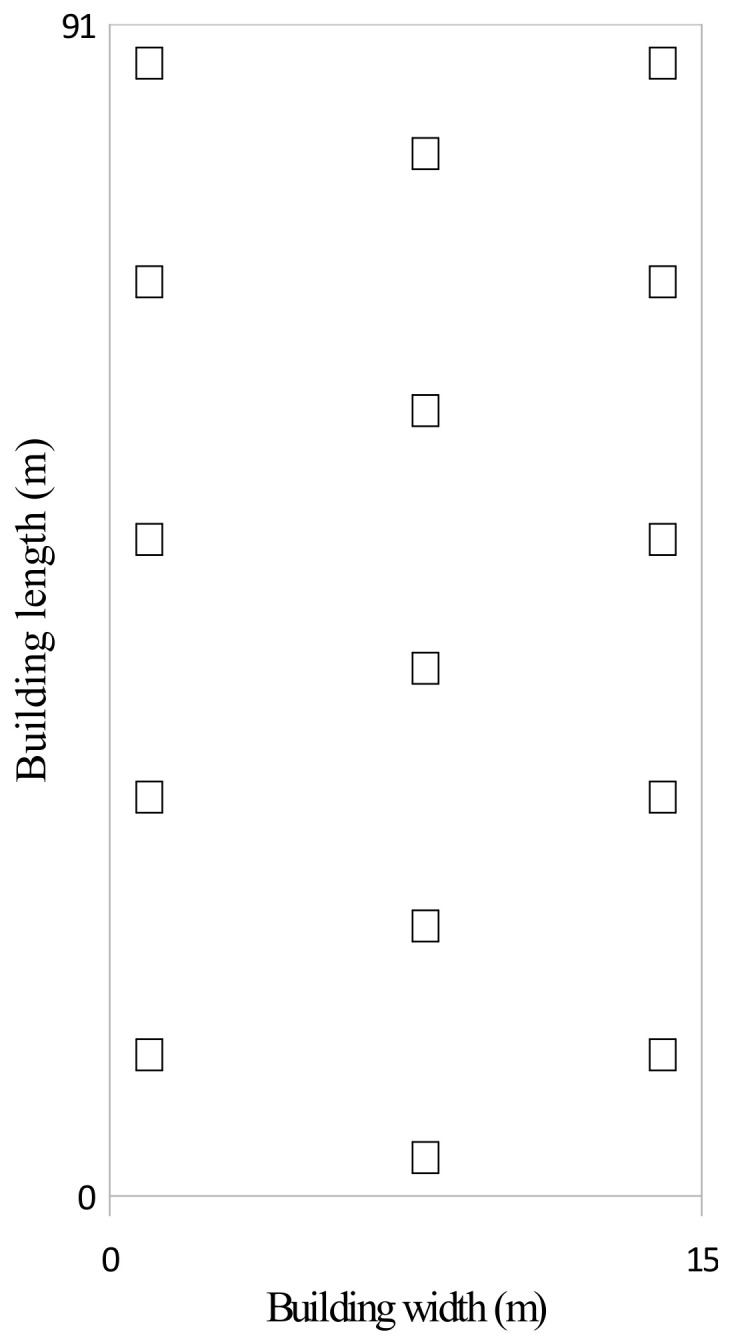
The distribution of insect traps in both buildings.

**Figure 2 animals-10-01461-f002:**
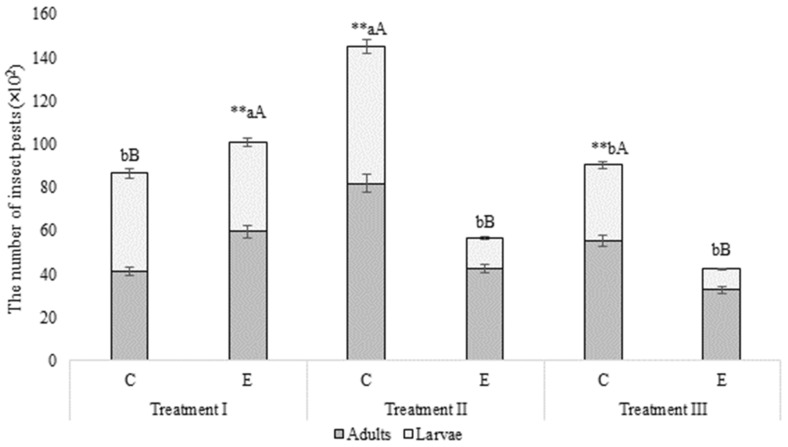
Number of larvae and adults in all treatments. ** Statistically significant differences (*p* < 0.05) in the number of insect between limewash and biocidal paint in every treatment; different lowercase in the number of larvae between limewash and biocidal paint in every treatment are significantly different (*p* < 0.05); different uppercase letters in the number of adults between limewash and biocidal paint in every treatment are significantly different (*p* < 0.05); Mann-Whitney U test was used; The standard error of measurement (SEM) is indicated on the bar.

**Table 1 animals-10-01461-t001:** Experimental scheme in laboratory bioassay.

Repetition	Applied Method	
	Limewash	Biocidal Paint
I	3 boxes (50 insects each)	3 boxes (50 insects each)
	3 boxes (50 insects each)	3 boxes (50 insects each)
	3 boxes (50 insects each)	3 boxes (50 insects each)
II	3 boxes (50 insects each)	3 boxes (50 insects each)
	3 boxes (50 insects each)	3 boxes (50 insects each)
	3 boxes (50 insects each)	3 boxes (50 insects each)
III	3 boxes (50 insects each)	3 boxes (50 insects each)
	3 boxes (50 insects each)	3 boxes (50 insects each)
	3 boxes (50 insects each)	3 boxes (50 insects each)

**Table 2 animals-10-01461-t002:** Dates of treatment.

Treatment	Length of Production Cycle (Arrival and Removal)	Group	
		C	E
I	22/12/16–30/01/17	15/12/16	15/12/2016
II	24/04/17–02/06/17	14/04/17	
III	02/01/18–10/02/18	23/12/17	

C: control group, E: experimental group.

**Table 3 animals-10-01461-t003:** Dates of sampling.

Treatment	Commercial Feed	Feeding	Date of Sampling
I	Starter	1–10 days	10th day of rearing—31/12/16
	Grower I	11–20 days	20th day of rearing—10/01/17
	Grower II	21–31 days	31st day of rearing—21/01/17
	Finisher	32–40 days	40th day of rearing—30/01/17
II	Starter	1–10 days	10th day of rearing—03/05/17
	Grower I	11–20 days	20th day of rearing—13/05/17
	Grower II	21–31 days	31st day of rearing—24/05/17
	Finisher	32–40 days	40th day of rearing—02/06/17
III	Starter	1–10 days	10th day of rearing—11/01/18
	Grower I	11–20 days	20th day of rearing—21/01/18
	Grower II	21–31 days	31st day of rearing—01/02/18
	Finisher	32–40 days	40th day of rearing—10/02/18

**Table 4 animals-10-01461-t004:** Mortality rates of *Alphitobius diaperinus* and mortality difference between larvae and adults (temperature and humidity: 22 °C ± 2 °C and 60–70%) in laboratories (n = 450).

Applied Method	Larval Mortality (%)				Adult Mortality (%)			
	7 d	10 d	14 d	22 d	7 d	10 d	14 d	22 d
Limewash	24.00 ^b^ (±1.18)	33.33 ^b^ (±1.38)	40.00 ^b **^ (±2.66)	49.33 ^b **^ (±3.28)	32.00 ^a **^ (±1.45)	34.67 ^b^ (±2.78)	37.33 ^b^ (±3.05)	42.67 ^b^ (±3.55)
Biocidal paint	41.33 ^aD **^ (±1.65)	58.67 ^aC **^ (±2.52)	82.67 ^aB **^ (±2.78)	100.00 ^aA^ (<0.001)	26.67 ^aD^ (±0.93)	44.00 ^aC^ (±2.31)	69.33 ^aB^ (±1.94)	100.00 ^aA^ (<0.001)
CV (%)		18.02		22.41				

± standard error of measurement (SEM); Values followed by the different lowercase letters in each column are significantly different (*p* < 0.05) according to the Student’s t-test, and followed by the different uppercase letters in each line are significantly different (*p* < 0.05) according to the Tukey’s multiple range test; ** Statistically significant differences (*p* < 0.05) between larvae and adults in limewash and biocidal paint in every treatment according to the Student’s test; CV: coefficient of variation.

**Table 5 animals-10-01461-t005:** Number of *Alphitobius diaperinus* per trap depends on the treatment (n = 60)**.**

Treatment	Group	Median	Min. Per Trap	Max. Per Trap	Percentual r/b
I	**C**	585 ± 35.22	0.00	1940	-
	**E**	670 ^a^ ± 35.07	0.00	1028	-
II	**C**	725 ± 71.62	0.00	3849	98.40 b
	**E**	350 ^b^ ± 22.52	0.00	793	22.86 r
III	**C**	590 ± 31.46	0.00	2987	53.97 b
	**E**	160 ^c^ ± 12.48	0.00	556	45.91 r

± standard error of measurement (SEM); The lowercase letters are significantly different from each other (*p* < 0.05) according to the Dunn’s test; C: control group, E: experimental group, r: reduction; b: boosting.

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
