# Peer review of "Effectiveness of Biocidal Paint Containing Permethrin, Ultramarine and Violet 23 Against Alphitobius diaperinus (Panzer) (Coleoptera: Tenebrionidae) in Laboratories and Poultry Houses"

_animals, 2020, doi:10.3390/ani10091461_

Round 1
Reviewer 1 Report
The study is very important. This insect is an important poultry production pest and a new technique to control it is very welcome. But is necessary point that the text has so many problems that are highlighted allong the text, in the comments.
INTRODUCTION: there is no justification to explain why the components of the ink and the ink itself were used.
METHODOLOGY: a negative control (without any product) was missing, both in the laboratory and in the aviary.
Larval mortality vs. adult mortality data were not analyzed (and the authors make comparative comments, but without the analysis it is wrong).
In lines 130-142 there is an exaggeration of unnecessary details, which do not interfere in the understanding of the study or in the Results.
RESULTS: there is a serious error in comparing the mortality of larvae x adults, without statistical basis. In Tables 4 and 5, data other than percentage values ​​should be removed. In addition, both should be brought together in a single table. In the presentation of the Results, there is an exaggerated use of words to explain, lack of objectivity.
DISCUSSION: the approach is superficial, and especially on lines 274-276, there is a problem regarding the effect of hydrated lime.
In lines 277-288 the text is confusing, very extensive (results are presented that apparently the authors consider their own, but which are related to the cited reference - 21). In addition, they use reference 21 a lot, which has methodological errors and wrong conclusions. It cannot be used.
In lines 289-299 the comparison with the use of fungus is mistaken, as in lines 300-311, the comparison with Bt is wrong. They are different modes of action. They cannot be related.
CONCLUSION: mistaken and very long (it is a summary of the main results).
In comment boxes are many others details

Author Response
Dear Reviewer 1,
We are very grateful for your response. We appreciate your comments. Your suggestions are very valuable. We are convinced that your insights will greatly improve the value of our manuscript. We responded to the Reviewer's comments and revised the manuscript as best we could.
THE TITLE was changed.
Line 20 – “livestock buildings for poultry” was changed into “poultry houses”
INTRODUCTION:
- We added a justification (now in lines 78-88).
METHODOLOGY:
- We did not use a negative group (without any product), as this would not be economically justified. The poultry producer would not have agreed to research in poultry houses without any insecticide. On the other hand, we wanted to do laboratory tests schematically related to poultry houses.
- All titles have been changed in accordance with the recommendation.
- Done (unnecessary details were removed).
- The website has been added (now in line 147).
RESULTS:
- We used the same titles as in Material and Methods.
- Text was rewritten (lines 179-192) and we did statistical analysis in order to compare larvae and adults as well as we used only percentual values (please, see Table 4).
- According to recommendation we changed “total abundance” on “the number of…” in all text.
- Text was rewritten – now lines 210-215.
- Text was rewritten – now lines 224-230.
- Table 5 – we added percentual reduction or boosting.
- We removed the mention of the behavior (in all text) due to unspecified parameters used in this observation.
DISCUSSION
According to recommendation we removed unnecessary information and references.
CONCLUSIONS:
We changed conclusions (now in lines 268-272).
All changes are marked in red in the text.
Reviewer 2 Report
The manuscript, "Effectiveness of biocidal paint containing permethrin,
ultramarine and violet 23 against Alphitobius diaperinus (Panzer) (Coleoptera: Tenebrionidae) in laboratory and production conditions", clearly demonstrated that biocidal paint could efficiently control the population of darkling beetle in chicken house, better than limewash and lasting longer. The authors employed both laboratory and production conditions to prove painting some biocidal is a good way to control beetles. Experimental design and the data collection are reasonable and believable. The results may be useful for chicken farmers to control the beetle.
Questions and changes:
- Figure 2: Treatment I, Treatment II and Treatment III replace I treatment, II treatment and III treatment.
- What is the initial population of beetles in production condition, very beginning date of starting chicken grower or the 10th day's sampling? Should mention in manuscript.
- Any different behavior of chicken between biocidal paint and limewash?
- Should discuss why the population was so high in treatment II. Perhaps due to the seasoning, such as temperature, moisture and sunshine, and etc.
Author Response
Dear Reviewer 2,
We are very grateful for your positive review and valuable comments, suggestions and questions. We are convinced that your insights will greatly improve the value of our manuscript.
Response 1. In the description of Figure 2 we changed ‘I treatment’, ‘II treatment’ and ‘III treatment’ into ‘Treatment I’, ‘Treatment II’ and ‘Treatment III’.
Response 2. Clarified in lines 148-150 and in lines 210-211 – the term ”initial population” referred to the population of the insect in treatment I. It has been corrected in the text to make it absolutely clear and not misleading.
Response 3. No changes in behavioral patterns were noted between chickens kept in the building painted with the biocidal paint and in the building with limewash. But we removed the mention of the behavior (in all text) due to unspecified parameters used in this observation.
Response 4. Now discussed in lines 211 – 215.
Round 2
Reviewer 1 Report
The text was improved than last version. But I recommend at "Results - poultry bioassay" (lines 200-216) a revision to make the text more clear and objective.

Author Response
Response to Reviewer 1 Comments
Dear Reviewer 1,
We are very grateful for your help.
- 2.3. Poultry bioassay – all suggestions have been taken into account (now in line 131-144 and 150).
- Line 174 – done (unnecessary details were removed).
- Line 175-191 - text was rewritten, all suggestions were included.
- Line 198 – we mentioned it (line 122-124).
- 3.2. Poultry bioassay – this subsection was rewritten (now line 199-217).
- Figure 2. – the tittle was improved; the indicated letters were corrected.
- Line 235 – done (now line 237).
- References – we used italic to scientific names.
All changes are marked in red in the text.